# Age-Dependent Differences in Postprandial Bile-Acid Metabolism and the Role of the Gut Microbiome

**DOI:** 10.3390/microorganisms12040764

**Published:** 2024-04-10

**Authors:** Soumia Majait, Emma C. E. Meessen, Mark Davids, Youssef Chahid, Steven W. Olde Damink, Frank G. Schaap, Ellis Marleen Kemper, Max Nieuwdorp, Maarten R. Soeters

**Affiliations:** 1Department of Pharmacy and Clinical Pharmacy, Amsterdam University Medical Center, 1105 AZ Amsterdam, The Netherlands; s.majait@amsterdamumc.nl (S.M.); y.chahid@amsterdamumc.nl (Y.C.); 2Department of Endocrinology and Metabolism, Amsterdam University Medical Center, 1105 AZ Amsterdam, The Netherlands; e.c.meessen@amsterdamumc.nl; 3Department of Internal and Vascular Medicine, Amsterdam University Medical Center, 1105 AZ Amsterdam, The Netherlands; m.davids@amsterdamumc.nl (M.D.); m.nieuwdorp@amsterdamumc.nl (M.N.); 4Department of Surgery, NUTRIM School of Nutrition and Translational Research in Metabolism, Maastricht University, 6229 ER Maastricht, The Netherlands; steven.oldedamink@maastrichtuniversity.nl (S.W.O.D.); frank.schaap@maastrichtuniversity.nl (F.G.S.); 5Department of General, Visceral and Transplantation Surgery, RWTH University Hospital Aachen, 52074 Aachen, Germany; 6Department of Experimental Vascular Medicine, Amsterdam University Medical Center, 1105 AZ Amsterdam, The Netherlands; e.m.kemper@amsterdamumc.nl

**Keywords:** anorexia of aging, sarcopenia, malnutrition, postprandial, bile acids, insulin resistance

## Abstract

Ageing changes the impact of nutrition, whereby inflammation has been suggested to play a role in age-related disabilities such as diabetes and cardiovascular disease. The aim of this study was to investigate differences in postprandial bile-acid response and its effect on energy metabolism between young and elderly people. Nine young, healthy men and nine elderly, healthy men underwent a liquid mixed-meal test. Postprandial bile-acid levels, insulin, glucose, GLP-1, C4, FGF19 and lipids were measured. Appetite, body composition, energy expenditure and gut microbiome were also measured. The elderly population showed lower glycine conjugated CDCA and UDCA levels and higher abundances of *Ruminiclostridium*, *Marvinbryantia* and *Catenibacterium*, but lower food intake, decreased fat free mass and increased cholesterol levels. Aging is associated with changes in postprandial bile-acid composition and microbiome, diminished hunger and changes in body composition and lipid levels. Further studies are needed to determine if these changes may contribute to malnutrition and sarcopenia in elderly.

## 1. Introduction

The worldwide population of those aged 65 years and older will be over 1.6 billion by 2050, leading to an increase in anorexia of ageing, sarcopenia and malnutrition [1].

Anorexia of ageing has been defined as an aged-related loss of appetite and/or decreased food intake. The prevalence of people with anorexia of ageing ranges from 25% in home dwellers to 62% in hospital populations and 85% in nursing-home populations. Consequences include sarcopenia, malnutrition, immunosuppression and frailty. This can ultimately lead to higher rates of morbidity and mortality. The main causes of anorexia of aging are thought to be changes in peripheral hormone signaling, gut motility and sensory perception [2].

Sarcopenia has been defined as an age-related loss of muscle mass and function. After the age of 50, muscle mass declines at a rate of 1% per year [3]. Malnutrition is a highly prevalent clinical condition in the elderly and has a major impact on morbidity and mortality [4]. The aetiology of malnutrition in the elderly is multifactorial with alternations in postprandial metabolism and the regulation of appetite thought to play important roles. Serra-prat et al. showed that elderly subjects had a less pronounced anabolic response compared to young, healthy controls. They found an enhanced postprandial cholecystokinin (CCK) and glucagon-like peptide (GLP-1) release in elderly subjects compared to controls. Postprandial hyperglycaemia and hyperinsulinemia were also found [5].

Ageing changes the impact of nutrition on a variety of metabolic events such as glucose and lipid metabolism and immune function. Low-grade inflammation has been suggested to play an important role in age-related disabilities such as obesity, diabetes and cardiovascular disease (CVD). Bile acids (BAs) are known for their role in hepatobiliary cholesterol secretion and lipid uptake via the gut. In recent decades, BAs have gained attention for their role as hormone-like mediators involved in energy metabolism [6]. During ageing, a decrease in bile-acid synthesis and a decrease in bacterial bile-acid modification occur [7]. Notably, BAs play a role in modulating postprandial glucose and lipid metabolism [8,9]. They activate the nuclear BA receptor farnesoid X receptor (FXR) for negative feedback control in which fibroblast growth factor 19 (FGF19) represses hepatic BA biosynthesis. BAs are ligands for the Takeda G-protein-coupled receptor 5 (TGR5) which enhances glucagon-like peptide-1 (GLP-1) secretion [8,10].

We hypothesised that the age-related decrease in anabolic response may be caused by diminished stimulating effects of BAs on meal-induced FGF19 and GLP-1 release in elderly compared to the young subjects. To that end, we explored the differences between elderly and younger subjects with respect to postprandial bile-acid response and its effect on energy metabolism. In addition, we investigated whether differences in gut microbiota composition could be linked to these outcomes.

## 2. Materials and Methods

### 2.1. Subjects

In this single-centre, observational study a total of eighteen participants were included: 9 healthy, young (18–30 y) male subjects and 9 healthy, elderly (>65 y) male subjects. Participants were included when they met the following criteria: (1) able to provide informed consent, (2) between 18 and 30 years at the time of signing the informed consent or 65 years or older, (3) body mass index (BMI) between 18.5 and 25 kg/m^2^, (4) Caucasian ethnicity, (5) general good health, as determined by medical history, physical examination by a physician and blood chemistry and (6) HOMA-IR index < 2.0. Exclusion criteria were: (1) major illness in the past 3 months, (2) use of any (self-)prescribed medication, (3) gastro-intestinal disease that may influence bile-acid metabolism, (4) history of cholecystectomy or other bile duct abnormalities, (5) tobacco smoking, (6) drug abuse, (7) alcoholism (>3 units a day), (8) serum creatinine > 120 µM, (9) fasting glucose ≥ 5.6 mmol/L, (10) ASAT > 80 U/L; ALAT > 90 U/L, GGT > 120 U/L and/or AP > 280 U/L (2 times upper limit reference interval) and (11) strenuous exercise for at least 3 days prior to each study day, defined as more than 1 h of exercise per day. To exclude the effect of postmenopausal insulin resistance on the anabolic response, we only included men. Written consent was obtained before the participants underwent one mixed-meal test and was in agreement with the principles of the Declaration of Helsinki (2013). The study was approved by the METC (Medical Ethics Committee, METC_2018_211) of Amsterdam UMC location AMC (AMC).

### 2.2. Mixed-Meal Test (MMT)

Participants visited the AMC at 08:00 after an overnight fast of 10 h. To achieve equally filled glycogen stores and a similar macronutrient balance, subjects avoided strenuous physical exercise and (excessive) alcohol intake for 72 h prior to the study days. Starting at 09:00 (T = 0), participants consumed a liquid mixed meal (Nutridrink Compact, Nutricia, Zoetermeer, The Netherlands) that contained 16% protein, 35% fat and 49% carbohydrates. Participants consumed the caloric equivalent of 25% of their daily resting energy expenditure (REE) measured with indirect calorimetry. During the mixed-meal test (MMT), arterialised blood was sampled from an IV cannula inserted in a dorsal hand vein or antecubital vein at 0, 30, 60, 90, 120, 150, 180, 210, 240, 270 and 300 min after consumption of the liquid mixed meal. At the start of the MMT, body composition was measured with whole-body air displacement plethysmography (Bodpod, Cosmed, Rome, Italy). Additionally, resting energy expenditure (REE) and respiratory quotient (RQ) were assessed with indirect calorimetry (Quark, Cosmed, Rome, Italy) using a ventilated hood system at baseline, 150 and 270 min. At the end of the MMT (5 h after liquid-meal intake) participants consumed an ad libitum lunch. The Sussex Ingestion Pattern Monitor (SIPM) was used to measure eating rate and volume of the meal. The SIPM consists of a concealed scale (Sartorius Cubis model), connected with a serial line to a laptop, and secured beneath a purpose-built table under a hole, on top of which a place mat was secured to allow positioning of the plate (SIPM, University of Sussex, Brighton, UK).

### 2.3. Laboratory Analysis

Blood samples obtained at the eligibility screening were analysed in the clinical laboratory of the AMC (LAKC). The total amount of blood drawn during the study was 132 mL, to determine the study parameters. Per MMT, a total of 10 blood samples were collected in the flowing tubes: heparin, EDTA, cloth and EDTA tubes, and centrifuged at 3000× *g* and 4 °C. Obtained sera/plasma was stored at −20 °C during the mixed-meal test day and at −80 °C until analysis. Plasma bile-acid levels and profile, FGF19 and 7α-hydroxy-4-cholesten-3-one (C4) were analysed at the Laboratory of the Department of Surgery at Maastricht University using an in-house-developed ELISA, as previously described [11]. Subjects were instructed to collect a morning faeces sample prior to the visit. This was directly frozen at −20°. Microbiota were analysed at the Microbiota Center Amsterdam using 16S rRNA sequencing. Plasma glucose levels were measured using the YSI 2500 Glucose Analyzer (Yellow Springs Instruments, Yellow Springs, OH, USA). Plasma insulin was measured by a chemiluminescence immunoassay (Siemens Healthcare Diagnostics, Breda, the Netherlands). Total plasma GLP-1 levels were measured by ELISA (Mercodia AB, Uppsala, Sweden).

### 2.4. Statistical Analysis

Data are presented as mean ± SD or median with range, depending on the distribution (tested by the Shapiro–Wilk test). Differences between and within groups were assessed with parametric and non-parametric tests, Levene’s test and Mann–Whitney U-test, respectively, depending on the distribution. Missing data of individual bile-acid concentrations below detection limit, were set as half the detection limit (0.5 nmol/L). SPSS Statistics 26 (IBM, Armonk, New York, NY, USA), GraphPad Prism version 8.30 (GraphPad Software Inc., La Jolla, CA, USA) and Rstudio were used for statistical analyses and graph design, respectively. A *p*-value ≤ 0.05 was considered statistically significant. Postprandial plasma levels were assessed as peak concentrations and area under the curve (AUC). Primary BAs were the sum of unconjugated and conjugated CDCA, CA and UDCA. Secondary BAs are the sum of unconjugated and conjugated DCA and LCA. Conjugated BAs were the sum of taurine and glycine conjugated Bas CDCA, CA, DCA, LCA and UDCA. Unconjugated BAs were the sum of CDCA, CA, DCA, LCA and UDCA. Associations with microbiome compositions were tested using permanova, while individual taxa were assessed by DESeq or Spearman correlation depending on the class of variable tested.

## 3. Results

### 3.1. Study Population

Twenty-eight subjects agreed to participate. Nine subjects did not meet the criteria. One subject stopped before the intervention. Therefore, a total of eighteen subjects were included. The study flow scheme is illustrated in Figure 1. In Table 1 the baseline characteristics of the study population are depicted. The elderly population showed significantly higher fasting plasma levels of glucose (*p* = 0.025), total cholesterol (*p* = 0.002), low-density lipoprotein cholesterol (LDL) (*p* < 0.001) and aspartate aminotransferase (ASAT) (*p* = 0.048) at screening.

### 3.2. No Differences in Postprandial Total Bile Acid, FGF19 or C4 Levels between Young and Elderly

In Figure 2, postprandial total bile acid (TBA), FGF19 and C4 levels are depicted. No significant differences were found between the two groups.

### 3.3. Individual BAs Levels Were Different between Young and Elderly

We measured total and individual BAs at baseline, and at three and five hours postprandially (Figure 3 and Table 2). In Figure 3 the average bile-acid composition at the three time points is depicted. The unconjugated CA fraction remained stable over time in the elderly population, whereas the fraction of glycine-conjugated CA increased between baseline and 3 h in the young population (*p* = 0.027) (Figure 3).

Glycine-conjugated CDCA (young: 2177 [1161, 4074] nmol/L vs. elderly: 1066 [718, 2007] nmol/L, *p* = 0.026) and UDCA (young: 368 [143, 579] nmol/L vs. elderly: 0.5 [0.5, 127] nmol/L, *p* = 0.015) were significantly higher in the young population 3 h after the meal. Also, significantly higher primary BAs were found in the young population (young: 1882 [1422, 3673] nmol/L vs. elderly: 436 [274, 2181] nmol/L, *p* = 0.040) and significantly higher conjugated BAs were found in the young population 5 h after the meal (young: 1776 [939, 3520] nmol/L vs. elderly: 602 [347, 933] nmol/L, *p* = 0.019) (Appendix A).

### 3.4. Microbiome Diversity Changed upon Ageing

Ageing is associated with increased α-diversity assessed by the Shannon index. Higher relative abundance of *Ruminiclostridium*, *Marvinbryantia* and *Catenibacterium* was found in the elderly group compared to the young population. Higher relative abundance of *Ruminococcaceae_UCG-004* and *Fusicatenbacter* was found in the young population compared to the elderly population (Figure 4).

In the overall population, we found associations at baseline between fecal microbiome composition and plasma UDCA, DCA, CDCA and CA. Also, *Slackia* seemed to be negatively associated with CDCA levels at baseline, while *Ruminococcaceae_UCG-013* was negatively associated with GCA after 5 h.

### 3.5. Healthy Ageing Did Not Affect Postprandial Glucose, Insulin or GLP-1 Levels

In Figure 5A–C, postprandial glucose, insulin and GLP-1 levels are depicted. HOMA-IR was calculated from fasting glucose and insulin levels (Figure 5D). No significant differences were observed.

### 3.6. Lipid Profiles Were Not Significantly Different between the Two Groups

Total cholesterol levels were significantly higher in the elderly compared to the young group at baseline (3.8 [3.2, 4.6] mmol/L vs. 5.9 [4.0, 6.5] mmol/L, *p* = 0.010), 2 h (3.7 [3.1, 4.5] mmol/L vs. 5.8 [3.8, 6.3] mmol/L, *p* = 0.011) and 4 h (3.7 [3.1, 4.6] mmol/L vs. 5.7 [3.9, 6.3] mmol/L, *p* = 0.010) after the start of the mixed-meal test (Figure 6A). No significant differences were found in baseline or postprandial triglyceride or HDL levels (Figure 6B,C).

### 3.7. The Elderly Men Ate Less and Had Lower Energy Expenditure

The young population consumed a significantly higher quantity of food during the ad lib meal compared to the elderly population (young: 844 ± 214 g/day vs. elderly: 575 ± 245 g/day, *p* = 0.049). Appetite and satiety were scored on a VAS scale. In both groups a significant difference in hunger before and after the meal was measured, see Appendix B. No differences in thirst, nausea or tastiness were found. The young population had significantly higher satiety scores after the meal compared to the elderly population. In both groups, satiety scores after the meal were significantly higher compared to baseline. A similar observation was found in satisfaction scores. Satisfaction scores after the meal were significantly higher compared to baseline. Differences between the two groups were not significant.

Energy expenditure data were measured for 20 min. Mean values were calculated from an interval between 5:00 and 15:00 min. Resting energy expenditure was significantly higher in the young population at t = 150 min (young: 2069 [1860, 2131] kcal/day vs. elderly: 1646 [1504, 1787] kcal/day, *p* = 0.03) (Figure 7A). When adjusted for bodyweight, resting energy expenditure was significantly higher in the young population at baseline (young: 24.4 [23.9, 25.8] kcal/kg BW/day vs. elderly: 22.0 [19.6, 24.3] kcal/kg BW/day, *p* = 0.013) and t = 150 min (young: 26.8 [25.4, 27.7] kcal/kg BW/day vs. elderly: 23.2 [21.0, 23.7] kcal/kg BW/day, *p* = 0.019) (Figure 7B). Respiratory quotient was significantly higher in the young population at baseline (young: 0.92 [0.87, 0.97] vs. elderly: 0.83 [0.82, 0.88], *p* = 0.029) (Figure 7C).

In Table 3 presents an overview of body composition measures. Significantly higher absolute fat mass and fat percentages were found in the elderly population compared to the young population. In addition, a significantly higher fat-free percentage was found in the young population compared to elderly population.

### 3.8. Diet

Participants reported their daily intake for three days prior to the visit days (Table 4). No difference in total daily caloric intake was found. The elderly population had a significantly higher vitamin B1 and vitamin B6 intake during dinner compared to the young population (See Appendix C).

## 4. Discussion

Aging is associated with changes in glucose, fat and protein metabolism and regulation of appetite. These changes are accompanied by loss of muscle mass. This also explains why elderly persons are prone to malnutrition that eventually may lead to higher morbidity and mortality rates [12]. As such, elderly subjects usually have a decreased anabolic response [3]. This phenomenon of age-related anabolic resistance was actually demonstrated in our study since we found higher fat mass and loss of appetite in the elderly subjects compared with the younger matched controls. The pathophysiology of this anabolic resistance in elderly people might be related to changes in bile acid and gut hormone secretion [5]. In this study we investigated age-related differences in postprandial bile acid and gut hormone secretion and its relation to the gut microbiome and other nutritional and metabolic factors.

Bile acids (BAs) are traditionally seen as lipid-emulsifying factors in the gut lumen; however, they also have endocrine-related functions by modulating postprandial glucose and lipid metabolism. They bind and activate FXR and are ligands for the TGR5 [8,10]. Both receptors are involved in metabolic and inflammatory control. BAs stimulate the production of GLP-1 via the TGR5. GLP-1, a gut hormone that is secreted after meal ingestion, potentiates glucose-stimulated insulin secretion and inhibits glucagon secretion in the pancreas. In addition, GLP-1 reduces gastrointestinal motility and appetite, whereby postprandial glucose excursions and food intake are kept down [13]. Another endocrine mediator whose synthesis is regulated by BAs is FGF19. FGF19 has been shown to downregulate CYP7A1, thereby decreasing bile-acid synthesis by a negative feedback loop [14]. It has also been shown to induce glycogen synthesis and regulate blood glucose and triglyceride levels. C4 is used as a serum marker for CYP7A1 enzymatic activity. The observed higher plasma levels of glycine-conjugated CDCA and UDCA in the young population were in line with previous studies [15] that have suggested that the postprandial conjugated bile-acid fraction in the portal blood is lower in elderly due to a less effective active ileal reabsorption of conjugated BAs in the elderly. The fact that GCDCA and GUDCA, specifically, were higher 3 h after the meal suggests a relationship between these two BAs. CDCA can be converted into UDCA upon bacterial epimerization of the C-7 hydroxyl group. UDCA formation from CDCA involves two enzymes: 7α-hydroxysteroid dehydrogenase (7α-HSDH) and 7β-hydroxysteroid dehydrogenase (7β-HSDH) [16].

We measured the individual bile-acid species only at T = 0, 3 and 5 h, but analysed the total BAs, FGF19 and C4 for the total postprandial response. This showed no differences between the groups. Hence, possible differences within the bile-acid composition may only be very small and could be due to baseline differences in the gut microbiome. The increased α-diversity that we found is not completely explained despite the fact that age-dependent changes in the gut microbiome composition have been described with some microbiota becoming more prevalent while others become less so [17]. However, mostly, a reduction in bacterial diversity was seen, which was attributed to age-related changes in the gastrointestinal tract, with a concomitant decline in immune function. One of the hypotheses for this loss in diversity during aging is that the opposite of neonatal gut colonization occurs, due to the changes described above, ultimately leading to an unstable microbiota composition and diversity loss. It could be the case that the elderly subjects in our study were relatively healthy, thereby not yet showing the initial stages of microbiota change. Otherwise, differences in microbiota analyses may contribute to the variability that is reported in the literature [17].

*Ruminiclostridium* is negatively associated with conjugated BAs [18] and a higher relative abundance is found in the elderly population. This was confirmed by our findings. Other studies have suggested that primary BAs are converted primarily into DCA in the elderly, rather than into LCA, due to slower colonic transit resulting in increased DCA absorption [19,20,21], which is not in line with our findings. Next to that, this finding could not be confirmed since no synthesis rates and/or colonic transit times were measured in this study. *Ruminococcaceae_UCG004* is known for the conversion of primary BAs to secondary BAs. *Ruminiclostridium* belongs to the family *Ruminococcaceae* [22]. The higher relative abundance of *Ruminococcaceae_ UCG004* in the young population was not in line with our data, suggesting a higher conversion of primary to secondary bile acid in the young population. *Marvinbryantia* is associated with obesity in rats, however our elderly population had a lean BMI [23]. No literature is available on associations between *Marvinbryantia* and bile-acid metabolism. *Catenobacterium mitsuokai P1-A4* contributes to the production of isolitocholic acid which is a bile acid formed from chenodeoxycholate by bacterial action [24]. We found a higher abundance of *Catenobacterium* in the elderly population but since we limited our analysis to the conjugated and unconjugated forms, this could not be further explained. Additionally, LCA is typically very low or absent in peripheral plasma samples [6,25]. Bacterial deconjugation is completed in the cecum and gut biotransformation of BAs (i.e., ratios of CA:DCA and CDCA:LCA) were altered between the two groups (Appendix A). This might suggest that the biosynthetic pathways of CA→DCA and CDCA→LCA were more active in the elderly compared to the young population and the role of the gut microbiome in it. It seems like a more diverse gut microbiome resulted in a greater conversion to secondary BAs.

Because we had found strong correlations and associations between GDCA, insulin and GLP-1 in previous studies, we expected GDCA to differ between the older and younger subjects, but this was not the case [6]. Even more, the groups did not differ in glucose, insulin or GLP-1 levels, which may be explained by the fact that we were able to match the groups quite well, thereby underestimating the insulin-resistant phenotype of older subjects. Another explanation may have been the small group size, but then, still no major effect was shown. Interestingly, we did show differences in lipid metabolism with a higher total cholesterol which might have been primarily driven by LDL cholesterol. Although human bile-acid metabolism is tightly coupled to cholesterol metabolism [26], it is unlikely that the higher cholesterol levels in our study were explained by differences in bile-acid biology.

Because we had a genuine interest in the human postprandial state we investigated eating behavior, energy expenditure and body composition. We confirmed earlier findings that elderly subjects eat smaller amounts when given a test meal. However, the ~32% lower meal intake of older subjects was not explained by differences in bile-acid-induced GLP-1 secretion which could have slowed down gastro-intestinal motility [27]. Notably, GLP-1 analogues are currently in widespread use as an anti-obesity strategy [28]. It should be noted that although the total amount of food intake was lower in the elderly group, perceived nausea scores did not differ between groups. Another feature of BAs may be the activation of energy expenditure via the TGR5, although the relevance for humans may be questioned [27,29]. We found higher energy expenditure in the younger subjects that, again, cannot be explained by bile-acid action. This increased energy expenditure is most likely explained by the differences in lean body mass since this contributes the most to daily caloric usage. Indeed, our data support the frequently encountered finding that older subjects have more fat mass at the cost of lean body mass. Finally, the nutritional intake records support the fact that the elderly subjects consumed less food, and carbohydrates in particular. In total this amounted to a difference of ~290 kcal per day. In particular, the small differences in the gut microbiome were not explained by differences in fiber intake and vegetables.

Our study had a few limitations, of which the small sample size was the most noticeable. In relation to bile-acid metabolism, this can be problematic due to inter-individual variability of the human postprandial bile-acid levels necessitating more study subjects. Even more, we have previously also shown intra-individual variability, highlighting this challenge [30]. The fact that inter-individual and intra-individual variability exist between different bile-acid species without clear effects on glucose and lipid metabolism may support that the total bile-acid levels outweigh individual BAs in terms of the TGR5 and/or FXR biological signaling relevance. Otherwise, bile-acid action outside the enterohepatic cycle may be of minor importance [25]. Another limitation was the fact that we only included male volunteers, although we did this on purpose to prevent the confounding effects of sex hormones and body composition [31]. Likewise, we included lean volunteers and our results may have been different for subjects with a more obese phenotype, which tends to be more common during aging. This study did not focus primarily on the mechanistic pathways which would have required not only interventions but possibly also in vitro studies. Finally, bile-acid levels in the peripheral circulation can be quite low and portal samples are only seldom available outside of surgical studies, which can hamper data interpretation.

In this study we aimed to explore the differences between elderly and younger subjects with respect to postprandial bile-acid response and gut microbiota and their effects on energy metabolism. Although we found subtle differences in individual bile-acid species and gut microbiota, these were not associated with differences in glucose, insulin or GLP-1. We deem it unlikely that bile-acid metabolism plays an important role in the metabolic changes that occur during aging.

## Figures and Tables

**Figure 1 microorganisms-12-00764-f001:**
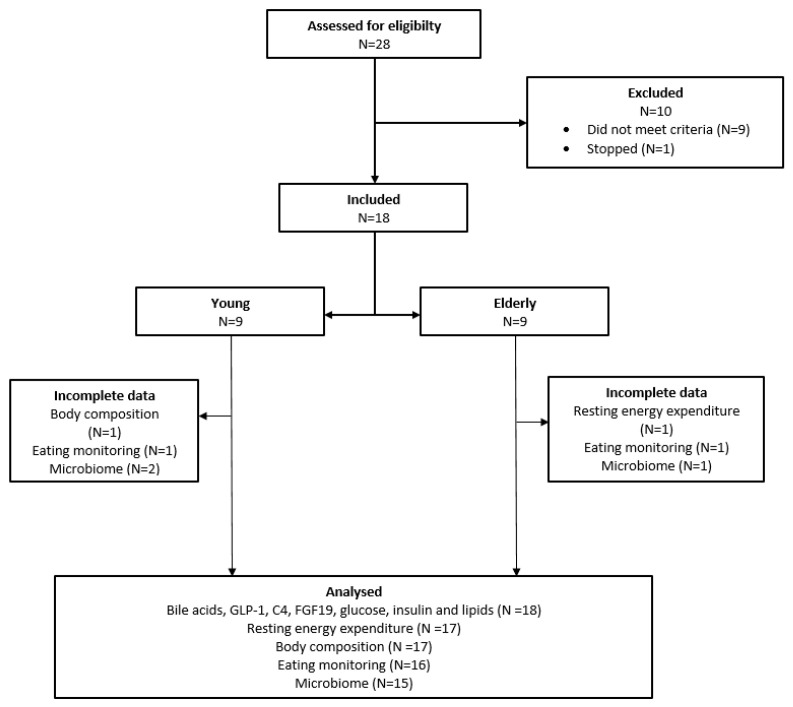
Study flow scheme.

**Figure 2 microorganisms-12-00764-f002:**
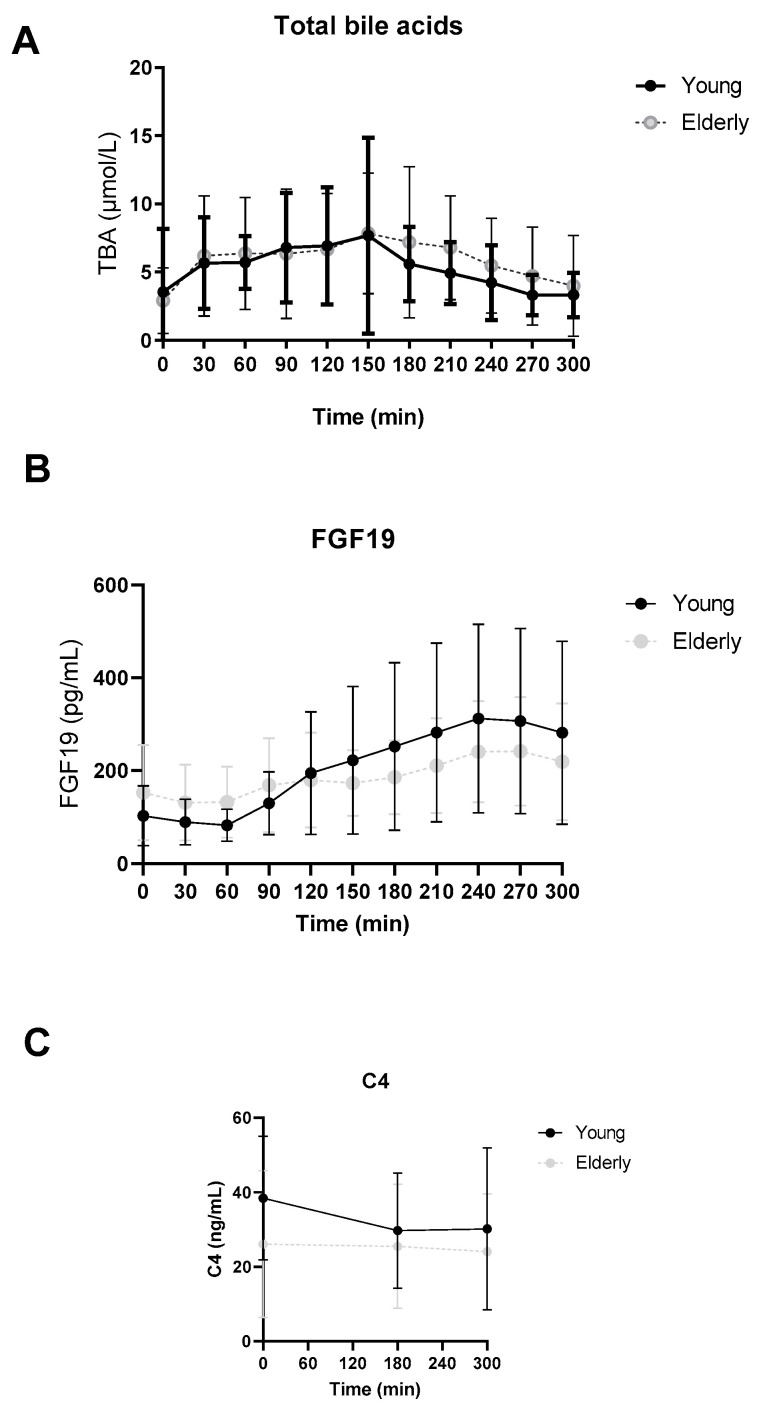
Postprandial total bile acid (**A**), FGF19 (**B**) and C4 (**C**) levels. ● young and ○ elderly men. Data are mean ± standard deviation (SD).

**Figure 3 microorganisms-12-00764-f003:**
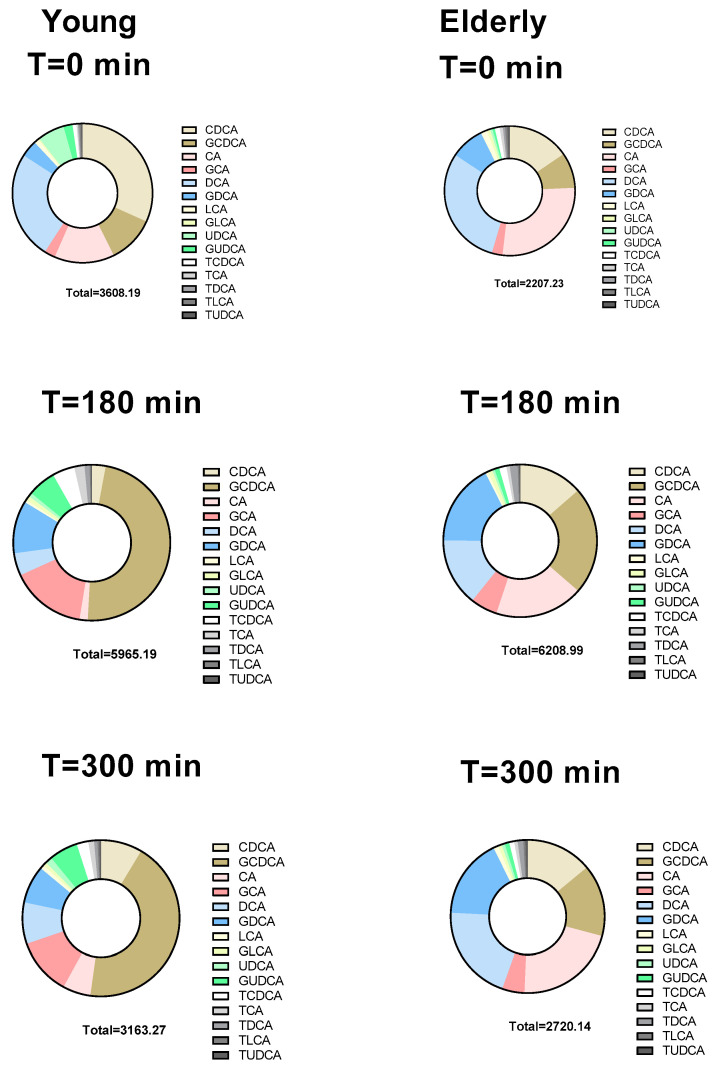
Fractional bile-acid composition in serum at baseline (T = 0), 3 h and 5 h postprandially. Total values are mean values of the sum of individual BAs (nmol/L). CDCA: chenodeoxycholic acid, CA: cholic acid, DCA: deoxycholic acid, LCA: lithocholic acid, UDCA: ursodeoxycholic acid, (G): glycine, (T): taurine.

**Figure 4 microorganisms-12-00764-f004:**
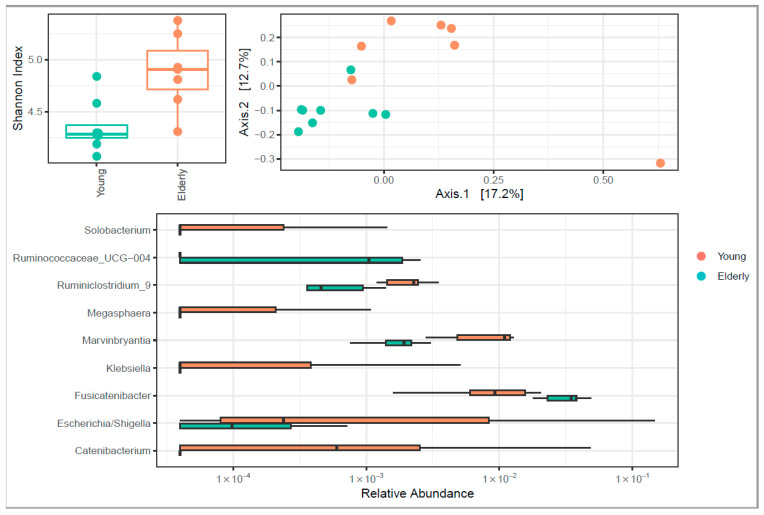
Relative abundance of fecal gut microbiota genus.

**Figure 5 microorganisms-12-00764-f005:**
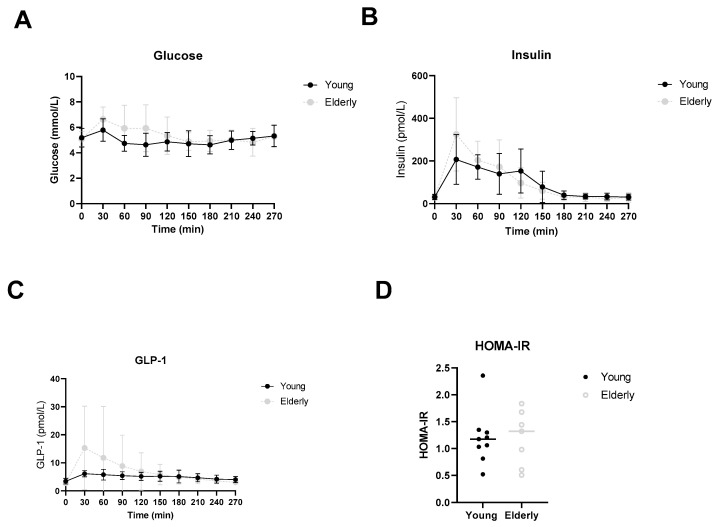
Postprandial glucose (**A**), insulin (**B**), GLP-1 (**C**) and HOMA-IR (**D**) levels. ● young and ○ elderly men. Data are mean ± standard deviation (SD).

**Figure 6 microorganisms-12-00764-f006:**
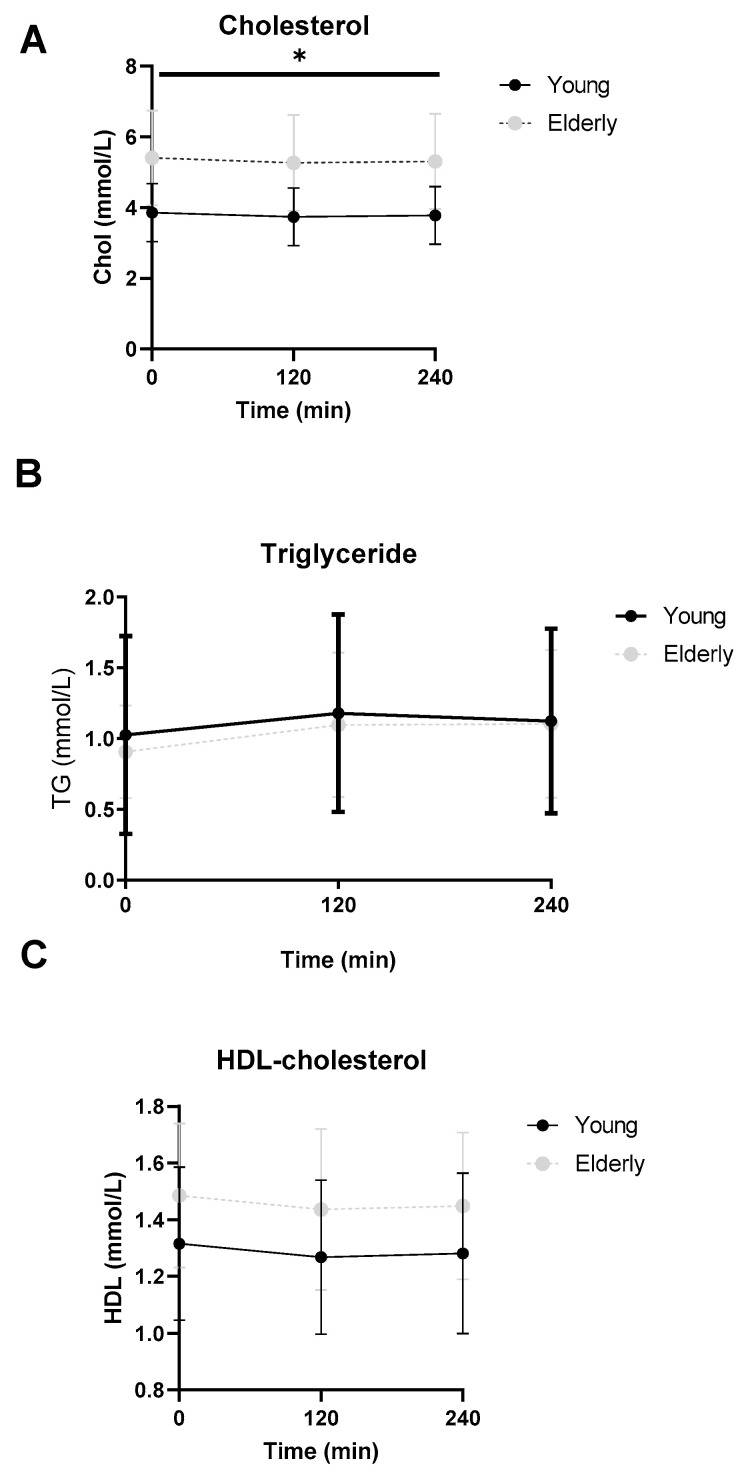
Postprandial cholesterol (**A**), triglyceride (**B**) and HDL cholesterol (**C**) levels. ● young and ○ elderly men. Data are mean ± standard deviation (SD). * at the curve represents a significant effect on the time point (*p* < 0.05).

**Figure 7 microorganisms-12-00764-f007:**
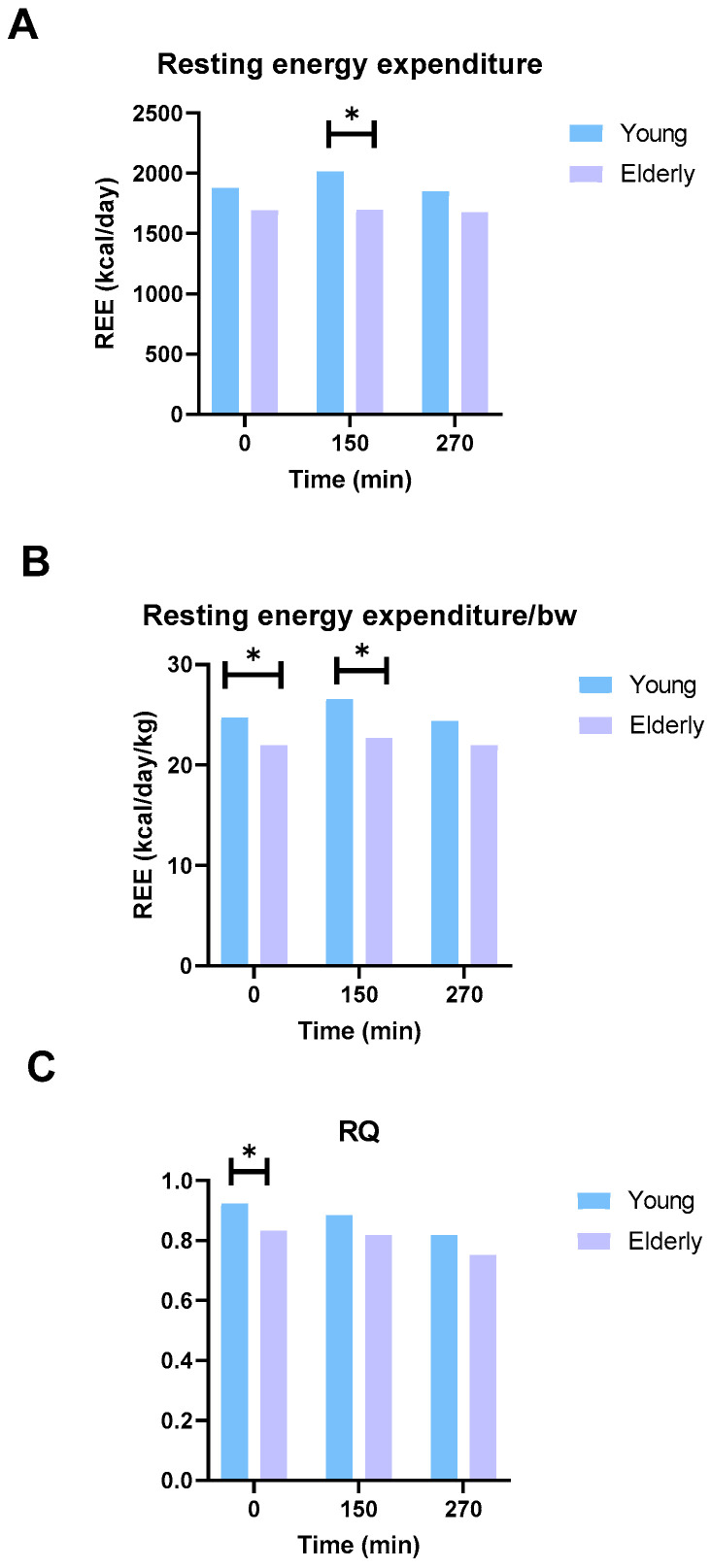
Average resting energy expenditure (**A**), resting energy expenditure adjusted for bodyweight (**B**) and respiratory quotient (**C**). * represents a significant difference (*p* < 0.05).

**Table 1 microorganisms-12-00764-t001:** Baseline characteristics.

	Young	Elderly	*p*-Value ^1^
**Age (years)**	24 [22, 26] ***	69 [66, 72] ***	<0.001
**Weight (kg)**	76.1 ± 6.8	79.0 ± 10.7	0.50
**Height (m)**	1.85 ± 0.07	1.85 ± 0.08	0.88
**BMI (kg/m^2^)**	22.2 ± 1.3	23.1 ± 2.4	0.32
**Creatinine (µmol/L)**	76 [74, 90]	81 [76, 88]	0.6
**EGFR (mL/min/1.73 m²) ^2^**	90 [90, 90]	86 [82, 90]	0.005
**AP (** **U/L)**	66 ± 15	72 ± 26	0.59
**GGT U/L)**	20 ± 10	28 ± 11	0.12
**ASAT (U/L)**	22 ± 4 *	27 ± 6 *	0.048
**ALAT (U/L)**	18 [17, 22]	29 [18, 42]	0.07
**Glucose (mmol/L)**	5.0 ± 0.2	5.2 ± 0.2	0.025
**Insulin (pmol/L)**	27.1 ± 10.5	26.6 ± 9.7	0.91
**HOMA-IR index**	0.94 [0.70, 1.4]	0.98 [0.75, 1.4]	0.79
**Total Cholesterol (mmol/L)**	3.77 ± 0.69 *	5.60 ± 1.27 *	0.002
**HDL (mmol/L)**	1.59 ± 0.41	1.55 ± 0.29	0.84
**LDL (mmol/L)**	1.79 ± 0.49 ***	3.64 ± 1.13 ***	<0.001
**Triglyceride (** **mmol/L)**	0.82 [0.47, 0.94]	0.80 [0.57, 1.3]	0.69

^1^ Data were assessed with Levene’s test when data were normally distributed and with Mann–Whitney U-test when data were not normally distributed. ^2^ EGFR: estimated glomerular filtration rate, AP: alkaline phosphatase, GGT: gamma-glutamyl transferase, ASAT: aspartate aminotransferase, ALAT: alanine aminotransferase, HOMA-IR: homeostatic model assessment for insulin resistance, HDL: high-density lipoprotein, LDL: low-density lipoprotein. * and *** at the curve represent a significant effect on the time point (*p* < 0.05, *p* < 0.01 and *p* < 0.001, respectively). Data are mean ± standard deviation or median [IR25%, IR75%].

**Table 2 microorganisms-12-00764-t002:** Individual serum bile-acid concentration at baseline, 3 h and 5 h postprandially.

	Baseline		3 h Postprandial		5 h Postprandial	
[BA] in nmol/L	Young	Elderly	Young	Elderly	Young	Elderly
**CDCA**	101 [591, 377]	0.5 ^a^ [0.5 ^a^, 390]	103 [84, 182]	56 [0.5 ^a^, 112]	233 [49, 348]	37 [0.5 ^a^, 336]
**CA**	105 [0.5 ^a^, 531]	0.5 ^a^ [0.5 ^a^, 551]	82 [23, 150]	18 [0.5 ^a^, 104]	60 [36, 289]	24 [0.5 ^a^, 281]
**DCA**	128 [0.5 ^a^, 560]	217 [68, 1024]	206 [0.5 ^a^, 489]	773 [72, 1596]	211 [0.5 ^a^, 385]	452 [46, 845]
**LCA**	32 [0.5 ^a^, 56]	31 [0.5 ^a^, 57]	30 [10, 43]	30 [0.5 ^a^, 58]	23 [0.5 ^a^, 37]	25 [0.5 ^a^, 40]
**UDCA**	0.5 ^a^ [0.5 ^a^, 70]	0.5 ^a^ [0.5 ^a^, 0.5 ^a^]	0.5 ^a^ [0.5 ^a^, 109]	0.5 ^a^[ 0.5 ^a^, 0.5 ^a^]	0.5 ^a^ [0.5 ^a^, 93]	0.5 ^a^ [0.5 ^a^, 0.5 ^a^]
**GCDCA**	309 [0.5 ^a^, 604]	0.5 ^a^ [0.5 ^a^, 220]	2177 [1161, 4074] *	1066 [718, 2007] *	891 [581, 2141]	274 [77, 521]
**GCA**	39 [0.5 ^a^, 176]	0.5 ^a^ [0.5 ^a^, 88]	381 [204, 1594]	218 [121, 508]	170 [60, 780]	79 [39, 148]
**GDCA**	0.5 ^a^ [0.5 ^a^, 258]	196 [0.5 ^a^, 271]	530 [352, 1052]	720 [363, 1474]	171 [41, 442]	218 [45, 426]
**GLCA**	0.5 ^a^ [0.5 ^a^, 0.5 ^a^]	0.5 ^a^ [0.5 ^a^, 15]	68 ± 44	70 ± 35	0.5 ^a^ [0.5 ^a^, 43]	0.5 ^a^ [0.5 ^a^, 0.5 ^a^]
**GUDCA**	0.5 ^a^ [0.5 ^a^, 154]	0.5 ^a^ [0.5 ^a^, 0.5 ^a^]	368 [143, 579] *	0.5 ^a^ [0.5 ^a^, 127] *	143 [69, 327]	0.5 ^a^ [0.5 ^a^, 43]

Data were assessed with Levene’s test when data were normally distributed and with Mann–Whitney U-test when data were not normally distributed. ^a^ values below LOD (<1 nmol/L) were set to LOD/2, resulting in a minimum concentration of 0.5 nmol/L. * at the curve represents a significant effect on the time point (*p* < 0.05). Data are mean ± standard deviation or median [IR25%, IR75%]. CDCA: chenodeoxycholic acid, CA: cholic acid, DCA: deoxycholic acid, LCA: lithocholic acid, UDCA: ursodeoxycholic acid, (G): glycine.

**Table 3 microorganisms-12-00764-t003:** Body composition.

	Young			Elderly			
Parameter	Mean	±	SD	Mean	±	SD	*p*-Value
**Fat mass (kg)**	7.32	±	2.79 ***	16.72	±	4.79 ***	<0.001
**Fat-free mass (kg)**	68.47	±	8.74	61.66	±	9.05	0.136
**Fat percentage (%)**	10.23	±	4.52 ***	21.32	±	5.03 ***	<0.001
**Fat-free percentage (%)**	90.14	±	4.31 ***	78.68	±	5.03 ***	<0.001
**Body weight (kg)**	75.82	±	7.21	78.38	±	10.45	0.571

Data were assessed with an unpaired *t*-test when data were normally distributed. *** represents a significant difference (*p* < 0.001). Data are mean ± standard deviation (SD).

**Table 4 microorganisms-12-00764-t004:** Dietary intake.

	Young			Elderly			
Dietary Constituent	Mean	±	SD	Mean	±	SD	*p*-Value
**Caloric intake (kcal/day)**	2178.0	±	659.1	1720.1	±	523.1	0.137
**Carbohydrates (g)**	262.9	±	94.1	189.8	±	58.7	0.078
**Fat (g)**	79.3	±	24.6	67.0	±	21.0	0.290
**Saturated fat (g)**	26.7	±	9.0	22.1	±	7.1	0.267
**Protein (g)**	87.1	±	35.9	74.7	±	32.3	0.466
**Dietary fibers (g)**	30.0	±	12.8	24.1	±	10.0	0.311

Data were assessed with an unpaired *t*-test when data were normally distributed. Data are mean ± standard deviation (SD).

## Data Availability

The data presented in this study are available on request from the corresponding author. The data are not publicly available due to privacy of the patients.

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
