# Peer review of "Age-Dependent Differences in Postprandial Bile-Acid Metabolism and the Role of the Gut Microbiome"

_microorganisms, 2024, doi:10.3390/microorganisms12040764_

Round 1

Reviewer 1 Report

Comments and Suggestions for Authors

The experimental study described in the paper focused on exploring differences in postprandial bile acid response and its effect on energy metabolism between young and elderly individuals. While the study investigated various aspects related to age-dependent changes in bile acid metabolism and the gut microbiome, it did not specifically delve into the mechanistic pathways. 

Why is there a difference in the microbiome study (2 for the young and 1 for the elderly)?

Use of ANY medication is a very broad exclusion criterion. Please specify.

Figure 8 should be in the supporting information.

The authors obtained very interesting and detailed numerical results. But unfortunately, the correlation analysis is missing. Individual measured factors are explained by age. However, no broader scientific discussion has been made. No analysis of BS vs cholestorol for example has been made. The effect of cholesterol and BS type on lipolysis may be hypothesized. The results show no discussion. No explanations are provided as to the justification for the observed change, and if it can be correlated to anything.

Author Response

REF 1

The experimental study described in the paper focused on exploring differences in postprandial bile acid response and its effect on energy metabolism between young and elderly individuals. While the study investigated various aspects related to age-dependent changes in bile acid metabolism and the gut microbiome, it did not specifically delve into the mechanistic pathways.

Author response

We thank the reviewer for the constructive remarks. We agree that this is not a paper that focuses primarily on the mechanistic pathways which would have would have required not only interventions but possibly also in vitro studies. We have mentioned this now in the limitations.

---

Why is there a difference in the microbiome study (2 for the young and 1 for the elderly)?

Author response

We think that the reviewer refers to the differences found in the microbiome (but we do not understand the 2 and 1 in the reviewers question). We have added more discussion now => 

The increased α-diversity that we found is not completely explained despite the fact that, changes in the gut microbiome composition have been described (12). One of the hypotheses is that during aging, the opposite of neonatal gut colonization occurs ultimately leading to an unstable microbiota composition. It could be the case that the elderly sub-jects in our study were relatively healthy thereby showing the initial stages of microbiota change. This change could then be explained by physiological changes in the gut, dietary patterns and the decline in immune function.

---

Use of ANY medication is a very broad exclusion criterion. Please specify.

Author response

We aimed for a healthy group and excluded all medically prescribed medication. We have added this to the methods section.

---

Figure 8 should be in the supporting information.

Author response

We have placed the figure in the appendix (B)

 ---

The authors obtained very interesting and detailed numerical results. But unfortunately, the correlation analysis is missing. Individual measured factors are explained by age. However, no broader scientific discussion has been made. No analysis of BS vs cholestorol for example has been made. The effect of cholesterol and BS type on lipolysis may be hypothesized. The results show no discussion. No explanations are provided as to the justification for the observed change, and if it can be correlated to anything

Author response

We agree with the reviewer that it would be interesting to have a more in depth discussion on results and their possible correlation. We have placed data in the appendix (A) that showed no correlation between HDL, TG, bile acids, FGF, C4, GLP1 and gut microbiota.

Additionally, one may be careful trying to discover correlations between LDL and other parameters because if a correlation was found, it might be based on the group difference in LDL and not on a true biological correlation per se. Also we did not measure lipolytic rates (FFA/glycerol levels or isotope flux).

Reviewer 2 Report

Comments and Suggestions for Authors

In general, the authors have produced a sound paper thoroughly investigating aging-related differences in cholesterol metabolism, BA syntheis and conjugation, and microbiota compoition, among other parameters tested by them.  However, (1) the sample population isof questionably small size, the authors should, therefore, be more modest in terms of their conculsions; (2) the reader might ask why the population selected by the authors is predominantly characterized by lean constitution, while a sizeable part of the human population is actually prone to developing obesity,  especially  asfar as the elderly are concerned: (3)  Otherwise, I consider the manuscript under reviewto be publishable 

Comments on the Quality of English Language

The text  needs editing in terms of language and spelling. to get rid of phrases like "are currently used widespread used".Different is an adjective, never a verb, among other similar minor language quibbles

Author Response

REF 2

In general, the authors have produced a sound paper thoroughly investigating aging-related differences in cholesterol metabolism, BA synthesis and conjugation, and microbiota composition, among other parameters tested by them.

However, (1) the sample population is of questionably small size, the authors should, therefore, be more modest in terms of their conclusions;

Author response

We also thank this reviewer for the constructive remarks. We agree with the reviewer on this point; hence we discussed this as limitation. We wonder if this is sufficient for the reviewer.

---

 (2) the reader might ask why the population selected by the authors is predominantly characterized by lean constitution, while a sizeable part of the human population is actually prone to developing obesity,  especially  as far as the elderly are concerned:

Author response

We aimed for “physiologically” healthy volunteers that only differed by age and not too much in body composition to prevent additional confounding of lean and fat mass. Nevertheless, there were age-related differences that could not be prevented. We have added in the discussion (limitations):

 Likewise, we included lean volunteers and our results may be different for subjects with a more obese phenotype which tends to be more common during aging.

---

Comments on the Quality of English Language

The text  needs editing in terms of language and spelling. to get rid of phrases like "are currently used widespread used".Different is an adjective, never a verb, among other similar minor language quibbles

Author response

We have changed these

Round 2

Reviewer 1 Report

Comments and Suggestions for Authors

The discussion and conclusion should be improved.

At the moment the authors analyze the age-related changes on the microbiome. However, the only hint at a justification of the change is the following sentences in the conclusion:

„One of the hypotheses is that during aging, the opposite of neonatal gut colonization occurs ultimately leading to an unstable microbiota composition. It could be the case that the elderly subjects in our study were relatively healthy thereby showing the initial stages of microbiota change.“

These sentences leave the reader unfortunately very unsatisfied. I encourage the authors to offer a more detailed description of what „opposite of neonatal gut colonization“, and why it would occur.

Again, a clear aim experimental aim, and hypothesis should be stated at the start. This has not been done. 

Author Response

Point 1: At the moment the authors analyze the age-related changes on the microbiome. However, the only hint at a justification of the change is the following sentences in the conclusion:

We have deepened the discussion that (indeed) was not clear enough:

The increased α-diversity that we found is not completely explained despite the fact that, age-dependent changes in the gut microbiome composition have been described with some microbiota becoming more prevalent while other do less so (17). However, mostly a reduction in bacterial diversity is seen, which is attributed to age-related changes in the gastrointestinal tract, with a concomitant decline in immune function. One of the hypotheses for this loss in diversity by aging is that the opposite of neonatal gut colonization occurs, due to the changes described above, ultimately leading to an unstable microbiota composition and diversity loss. It could be the case that the elderly subjects in our study were relatively healthy thereby not yet showing the initial stages of microbiota change. Otherwise, differences in microbiota analyses may contribute to the variability that is reported in the literature (17).

Point 2: Again, a clear aim experimental aim, and hypothesis should be stated at the start. This has not been done.

We clarified the hypothesis and aims in the last paragraph of the introduction:

We hypothesize that the age related decrease in anabolic response may be caused by diminished stimulating effects of BAs on meal-induced FGF19 and GLP-1 release in elderly compared to the young subjects. To that end, we explored the differences between elderly and younger subjects with respect to postprandial bile acid response and its effect on energy metabolism. Additionally we investigated whether differences in gut microbiota composition can be linked to these outcomes.